# Robustly measuring multimorbidity using disparate linked datasets
Regina Prigge [1] ✉, Kelly J. Fleetwood [1], Caroline A. Jackson [1], Stewart W. Mercer[1],
Paul AT Kelly [2], Cathie Sudlow [1,3], John D. Norrie[1], Daniel R. Morales[4,5], Daniel J. Smith[6] &
Bruce Guthrie [7]

## Abstract

**Background** Measurement of multimorbidity, the co-occurrence of two or more conditions in the same individual, is highly variable which limits the consistency and reproducibility of research.
**Methods** Using data from 172,563 UK Biobank (UKB) participants and a cross-sectional approach, we examined how choice of data source affected estimated prevalence of 80 individual long-term conditions (LTCs) and multimorbidity. We developed code-list-based algorithms to determine the prevalence of 80 LTCs in (1) primary care records, (2) UKB baseline assessment, (3) hospital/cancer registry records, and (4) all three data sources together.
**Results** Using records from all three data sources, 146,811 (85.1%) participants have at least one and 109,609 (63.5%) have at least two LTCs at baseline. A median of 4.7% (IQR 1.0–16.6) of participants with a condition are identified by all three data sources. Agreement is highest for endocrine, nutritional and metabolic disorders, with a median of 32.9% (IQR 20.5–34.1) of individuals with a condition identified by all three data sources. Agreement is lowest for diseases of the genitourinary system and mental and behavioural disorders where perfect agreement varies from zero to 4.9% and zero to 12.3% across conditions, respectively. The low agreement between data sources is accompanied by high proportions of individuals with a condition identified only in primary care data (i.e. not in either of the other two sources), with a median of 59.3% (IQR 47.4–75.9) for diseases of the genitourinary system and 66.9% (IQR 42.8–79.2) for mental and behavioural disorders.
**Conclusions** Our study highlights the impact of the choice of which data source is used in research on individual LTCs and multimorbidity, and the importance of clearly justifying choices made.

## Plain language summary

Researchers with an interest in the co-occurrence of multiple long-term health conditions can use different sources of information to identify these conditions. For example, when people are seen in primary care practices or hospitals, doctors will record their diagnoses in computer records. Furthermore, if people agree to participate in research studies, they may self-report their conditions. We explored how the choice of which data source is used affects the estimated number of people with 80 individual and multiple long-term conditions. Our study highlights that choice of data source has major implications on the number of people with health conditions identified, and for many conditions there is poor agreement between data sources. Researchers should explicitly state and justify their choices to improve reproducibility of research.

The co-occurrence of multiple long-term health conditions (MLTCs or multimorbidity) is a public health concern because it is both common and associated with increased health care utilisation and risk of mortality[1,2]. Despite its importance, there is huge variation in the measurement of multimorbidity in research, in terms of both the number of conditions and the specific conditions included[3]. Furthermore, the measurement of individual conditions is inconsistent and poorly reported, making reproducibility and comparability of existing analyses challenging[3].

Large prospective cohort studies such as the UK Biobank (UKB) have created unprecedented opportunities for multimorbidity research due to the wealth of information collected on a broad range of physical and mental health conditions[4]. UKB collected information from participants through questionnaires, interviews and collection of physical measures and samples at baseline. In addition, there is linkage to electronic health records which allows the identification of health conditions in the period up to recruitment and during follow-up.

[1]Usher Institute, University of Edinburgh, Edinburgh, UK. [2]Public Member of Study Advisory Board, Edinburgh, UK. [3]British Heart Foundation Data Science Centre, Health Data Research UK, London, UK. [4]Division of Population Health and Genomics, University of Dundee, Dundee, UK. [5]Department of Public Health, University of Southern Denmark, Odense, Denmark. [6]Centre for Clinical Brain Sciences, University of Edinburgh, Edinburgh, UK. [7]Advanced Care Research Centre, Usher Institute, University of Edinburgh, Edinburgh, UK. ✉e-mail: regina.prigge@ed.ac.uk

Multimorbidity research in UKB to date has typically used bespoke data collected at the UKB baseline visit and/or data from linked hospital and death records[5–7]. Far less attention has been paid to linked primary care records, perhaps because primary care data has only been available for a subset of participants or because of the complexity of identifying health conditions in primary care records. However, many important conditions, including common mental health conditions, do not commonly result in a hospital admission, meaning that they are under-ascertained when only using hospital data[8]. All researchers face a number of choices in measuring multimorbidity, namely: (1) Which long-term conditions (LTCs) to include in multimorbidity measures? (2) Which datasets and code-lists to use to measure chosen LTCs? and (3) Which measure of multimorbidity to use? Choices should be appropriate to the purpose of each study, but are often not explicitly stated or justified, which particularly applies to choice of data source[3,9,10].

To advance our understanding of the impact of different choices of data source, our aim is to determine the prevalence of 80 individual LTCs and of multimorbidity in UKB based on ascertainment from different data sources, and using four different definitions of multimorbidity. To further support the efficiency and reproducibility of future research involving primary care records, we additionally provide researchers with a comprehensive set of Read v2 and Clinical Terms Version 3 (CTV3) code-lists to ascertain LTCs in primary care records.

We find that choice of data source has major implications on ascertainment of most conditions, and for many conditions there is poor agreement between data sources. Compared to using all three data sources to identify conditions, primary care records alone identify the highest proportion for 62 of the 80 conditions, compared to 13 conditions for UKB baseline assessment data alone and only five rare conditions for hospital records alone. We also find that estimates of MLTC prevalence vary greatly depending on the definition of multimorbidity and depending on the data sources used, with estimates based on primary care data alone higher than those based on other data sources alone and with hospital ascertainment of physical–mental multimorbidity particularly deficient.

## Methods
### Study design and participants
Our study population included UKB participants aged 40–71 years at baseline, recruited in 22 assessment centres in England, Scotland, and Wales from 2006 to 2010, with a continuous primary care record extending from at least a year prior to and up to the day of their baseline assessment. Linked primary care data was available from Scotland, Wales and practices in England that used either the TPP or Vision practice management systems. We excluded primary care records from the UKB extract of the Vision practice management system in England because this linked dataset did not include participants who died before data extraction[11]. Analyses of UKB data are conducted under generic approval from the NHS North West Research Ethics Service (Ref. 21/NW/0157) based on individual written informed consent at baseline assessment. Participants who subsequently withdrew consent were excluded from the analysis.

### Data sources
We used information from four data sources: (1) UKB baseline assessment data (Supplementary Data 1)[4]; (2) hospital admission records coded with ICD-10 for diagnoses and OPCS-4 for procedures[12]; (3) cancer registry; and (4) primary care records. UKB participants gave consent for their information to be linked with electronic health records. We collated information stored in these health records based on tables provided by UKB. The first three data sources are complete for every participant but at the time of analysis linkage to primary care records was only available for approximately half of participants. We defined conditions at baseline using all primary care records up to and including the date of the participant's baseline assessment. Cancer registry and hospital records were available from different dates for England, Wales and Scotland, with a minimum of eight years of records prior to the baseline assessments (Supplementary

Table 1). To maximise the use of available data whilst ensuring consistency across the secondary care data sources and countries, we therefore defined conditions at baseline using cancer registry and hospital records from the eight years up to and including their baseline assessment date.

### Definition of variables
**Individual LTCs.** Three clinicians (B.G., S.W.M., D.J.S.) selected LTCs to include from a previously published set of 308 acute and chronic conditions[13], with discussion in case of disagreement (Supplementary Methods 1). In short, each clinician was asked to select conditions that were: (1) a long-term physical or mental health condition and (2) of relevance to middle-aged adults (the recruited population in UKB). Based on clinical consensus, 78 conditions were defined, and we subsequently added 2 conditions (post-traumatic stress disorder and Addison's disease) recommended by a Delphi consensus study of conditions that should always be included in multimorbidity research[9]. Eighty conditions were therefore ascertained, grouped by body systems in keeping with ICD-10 chapters and categorised as mental or physical LTCs (Supplementary Data 2).

UKB baseline assessment data included information from a touchscreen questionnaire, nurse interview, physical and functional measures, and collection of blood, urine, and saliva[4]. We identified relevant information on the presence of 72 conditions, which was largely based on self-report in the touchscreen questionnaire and nurse interview but also included laboratory values of creatinine to ascertain chronic kidney disease (Supplementary Data 1).

We identified records of each condition in electronic health records using existing and newly developed code-lists which are available in our GitHub repository[14]. https://github.com/rprigge-uoe/mltc-codelists.git (Supplementary Data 3). Using primary care data is not straightforward, since data are extracted from three electronic information systems (EMIS[15], Vision[11], and SystmOne[16]) using two different coding systems (Read v2 and CTV3). We briefly summarise key challenges and how we addressed them below (detailed methods in Supplementary Methods 2). Where possible, we used existing and validated Read v2, CTV3, ICD-10 and OPCS-4 code-lists, largely created by researchers using the CALIBER platform[13,17] but also code-lists identified via the OpenSafely repository[18], on the UKB website[19], by contacting topic experts, and through literature review. Most published code-lists were for Read v2 and ICD-10, meaning that we often had to create new CTV3 code-lists. For Read v2, UKB (and some other data providers such as SAIL Databank) only provide the first five digits of the full seven-digit codes. Most published code-lists provide the full seven-digit codes, and whilst the first five-digits are usually sufficient to specify conditions, there are situations where the full seven-digit code is needed. Hence, we had to modify published Read v2 code-lists to be suitable for these data. In addition, we modified some existing code-lists to be more appropriate for our purpose (Supplementary Table 2). For example, the CALIBER code-list for 'hearing impairment is very broad, but we chose to focus more narrowly on bilateral moderate or severe deafness and/or evidence of hearing aid use. Where there was no existing code-list (e.g. Addison's disease), we created new code-lists.

For each participant, we identified the first record of each condition in each data source up to their UKB baseline assessment date. The diagnosis date of most conditions was defined as the earliest record of the condition, but eight lifelong conditions (autism and Asperger's syndrome, intellectual disability, Down's syndrome, cerebral palsy, sickle-cell anaemia, thalassaemia, cystic fibrosis, juvenile arthritis) were defined as present from birth. Additional rules were developed to deal with overlaps across mutually exclusive conditions (e.g. type 1 diabetes, type 2 diabetes, and diabetes not otherwise specified) or other exceptions (e.g. sex-specific conditions) (Supplementary Table 3).

**Multimorbidity.** We identified participants with multimorbidity based on four definitions[20]: (1) MLTC 2+ was defined as the presence of at least two of the 80 LTCs which is the recommended core definition of multimorbidity[9,21]; (2) MLTC 3+ was defined as the presence of at least

three LTCs; (3) MLTC 3+ from 3+ was defined as the presence of at least three LTCs from at least three different body systems; and (4) 'Mental–physical multimorbidity' was defined as the presence of at least two LTCs, where at least one was a mental health LTC and at least one was a physical health LTC.

**Covariates.** Age at baseline and sex were ascertained from recruitment data. Participants self-reported their ethnicity in the touch-screen questionnaire. We categorised ethnicity into two groups (White; ethnic minority groups). Area-based deprivation, measured by the Townsend Deprivation Index, was derived from participants' home addresses at baseline, and was categorised into deciles within the entire UKB cohort.

### Statistics and reproducibility
We report baseline characteristics for the whole population as well as for subsets of the population meeting each multimorbidity definition using data from (1) primary care records alone; (2) UKB baseline assessment records alone (hereafter referred to as UKB records alone), (3) hospital admission and/or cancer registry records alone (hereafter referred to as hospital records alone), and (4) all three data sources together. For each individual LTC and data source, we report the number and proportion of the study population with the condition, and the degree of concordance between data sources. In addition, we determined the number and proportion of the study population meeting each multimorbidity definition overall and by age, sex, ethnicity and deprivation using data from all data sources and each data source alone. We performed available case analyses since age and sex were fully observed and ethnicity and deprivation were each missing for less than 1% of participants.

### Reporting summary
Further information on research design is available in the Nature Portfolio Reporting Summary linked to this article.

## Results
### Study population
The study population includes 172,563 UKB participants (Supplementary Fig. 3). At baseline, participant mean (SD) age is 56.7 (8.0) years, 94,029 (54.5%) are women, and 164,711 (95.4%) have a white ethnic background (Table 1). Using data from all three data sources, 109,609 (63.5%) have MLTC 2+, 74,283 (43.0%) have MLTC 3+, 61,902 (35.9%) have MLTC 3+ from 3+ and 38,411 (22.3%) have mental–physical multimorbidity at baseline (Table 1). Compared to people with multimorbidity under other definitions, participants with mental–physical multimorbidity are younger at baseline, are more commonly women or white, and are more likely to live in socioeconomically deprived areas.

### Individual LTCs
Using records from all three data sources, 146,811 (85.1%) participants have at least 1 of the 80 LTCs at baseline (Supplementary Data 4). The three most prevalent conditions are hypertension ($n = 52,257$, 30.3%), allergic and chronic rhinitis ($n = 47,830$, 27.7%), and osteoarthritis ($n = 35,417$, 20.5%). The three least frequent conditions are Down's syndrome ($n = 24$, 0.01%), motor neurone disease ($n = 37$, 0.02%), and sickle-cell anaemia ($n = 53$, 0.03%). Using records from one data source at a time, the proportion of participants with a condition identified in each of primary care records, UKB records and hospital records alone greatly differ across conditions. The proportion of participants with a condition identified range from 27.5% to 99.6% in primary care records alone, from 0% to 99.3% in UKB records alone and from 0.4% to 73.9% in hospital records alone. For 62 of the 80 conditions, primary care records identify the largest proportion out of the three data sources. UKB records identify the largest proportion for 13 (mostly common) conditions, whereas hospital records identify the largest proportion for only 5 (all rare) conditions (bold numbers in Supplementary Data 4).

Concordance between data sources varies across conditions and body systems (Fig. 1, Supplementary Fig. 4, and Supplementary Data 5). The proportion of participants identified as having a condition by all three data sources ranges from 0% for nine conditions to 55.1% for type 1 diabetes. A median of 4.7% (interquartile range (IQR) 1.0–16.6) of individuals with a condition are identified by all 3 data sources, and only 7 conditions have more than one-third of participants with the condition identified in all 3 data sources (type 1 diabetes, solid organ malignancies, haematological malignancies, coronary heart disease, multiple sclerosis, type 2 diabetes and Addison's disease). Agreement is highest for endocrine, nutritional and metabolic disorders, with a median of 32.9% (IQR 20.5–34.1) of individuals with a condition identified by all three data sources, followed by neoplasms with a median of 28.5% (IQR 13.3–41.8). Agreement is notably poor for diseases of the genitourinary system and mental and behavioural disorders where perfect agreement varies from zero to 4.9% and zero to 12.3% across conditions, respectively. For both body systems, this is accompanied by high proportions of individuals with a condition identified only in primary care data (i.e. not in either of the other two sources), with a median of 59.3% (IQR 47.4–75.9) for diseases of the genitourinary system, and 66.9% (IQR 42.8–79.2) for mental and behavioural disorders. Focusing on only two data sources at a time, a median of 22.9% (IQR 8.3–41.7) of individuals with a condition are identified by primary care and UKB records, 17.3% (IQR 6.5–26.7) by primary care and hospital records and 10.8% (IQR 5.2–23.7) by UKB and hospital records (Supplementary Figs. 5–7).

### Multimorbidity
Using data from all three data sources, prevalence of MLTC 2+, MLTC 3+ and MLTC 3+ from 3+ all rise steadily with age, but prevalence of physical–mental multimorbidity is fairly constant (Fig. 2, Supplementary Fig. 8, and Supplementary Data 6). Women have higher prevalence of all types of multimorbidity but sex differences reduce with age. People from minority ethnic groups have lower prevalence of physical–mental multi-morbidity, but higher prevalence of all three other types with widening gaps with increasing age. People living in the most deprived areas have higher prevalence of all types of multimorbidity at all ages.

The estimated prevalence of multimorbidity measured in each of the four ways varies considerably between data sources (Table 1 and Supplementary Tables 4–6), with estimates ranging from 12.0% to 50.9% for MLTC 2+, from 5.5% to 31.3% for MLTC 3+, from 3.5% to 25.2% for MLTC 3+ from 3+, and from 1.1% to 18.8% for mental–physical multi-morbidity. While the characteristics of the population of people identified as having multimorbidity are similar in analyses using primary care records alone and UKB records alone, the population of people with multimorbidity identified using hospital records alone are slightly older, include fewer women, and have a higher proportion of ethnic minority groups and people living in the most deprived areas.

Irrespective of the multimorbidity definition, multimorbidity prevalence measured using a single data source is highest based on analyses using primary care records alone, followed by UKB records alone, and lowest using hospital records alone (Fig. 3 and Supplementary Data 7). Broad patterns of changing prevalence with age are similar irrespective of which data source is used to identify conditions. For physical–mental multimorbidity, the relationship with age is fairly flat, with a slight increase in the prevalence up to ages 55–59 years then a small decrease in older age groups. For the other three definitions of multimorbidity, prevalence rises steadily with age.

Irrespective of the multimorbidity definition, patterns of multi-morbidity prevalence stratified by sex, ethnicity and deprivation measured using a single data source are similar to those seen in analyses using all three data sources together (Supplementary Data 6, 8–10). The exception is that prevalence of multimorbidity is higher among men than women in analyses using hospital records alone, and sex differences are smaller in analyses using UKB records alone than in analyses using primary care records alone or using all data sources together.

**Table 1 | Baseline characteristics of all eligible UK Biobank participants and for subsets of the study population meeting each of the four multimorbidity definitions (multimorbidity ascertained using all three data sources)**

| | Eligible participants | People with MLTC 2+ | People with MLTC 3+ | People with MLTC 3+ from 3+ | People with mental–physical multimorbidity |
|---|---|---|---|---|---|
| N (% of whole population) | 172,563 (100) | 109,609 (63.5) | 74,283 (43.0) | 61,902 (35.9) | 38,411 (22.3) |
| **Age group, years, (%)** | | | | | |
| 40–44 | 16,640 (9.6) | 7577 (6.9) | 4104 (5.5) | 2894 (4.7) | 3280 (8.5) |
| 45–49 | 22,351 (13.0) | 11,157 (10.2) | 6284 (8.5) | 4758 (7.7) | 4522 (11.8) |
| 50–54 | 26,130 (15.1) | 14,841 (13.5) | 9269 (12.5) | 7454 (12.0) | 5947 (15.5) |
| 55–59 | 31,513 (18.3) | 19,912 (18.2) | 13,332 (17.9) | 11,161 (18.0) | 7445 (19.4) |
| 60–64 | 42,372 (24.6) | 29,818 (27.2) | 21,217 (28.6) | 18,122 (29.3) | 9850 (25.6) |
| 65+ | 33,557 (19.4) | 26,304 (24.0) | 20,077 (27.0) | 17,513 (28.3) | 7367 (19.2) |
| Women (%) | 94,029 (54.5) | 60,943 (55.6) | 41,415 (55.8) | 35,223 (56.9) | 24,055 (62.6) |
| **Ethnicity (%)** | | | | | |
| White | 164,711 (95.4) | 104,626 (95.5) | 70,836 (95.4) | 59,019 (95.3) | 36,935 (96.2) |
| Ethnic minority groups | 7109 (4.1) | 4489 (4.1) | 3104 (4.2) | 2591 (4.2) | 1309 (3.4) |
| Missing | 743 (0.4) | 494 (0.5) | 343 (0.5) | 292 (0.5) | 167 (0.4) |
| **Area-based deprivation (deciles), (%)[a]** | | | | | |
| 1 (least deprived) | 17,003 (9.9) | 10,292 (9.4) | 6618 (8.9) | 5414 (8.7) | 3260 (8.5) |
| 2 | 18,230 (10.6) | 11,169 (10.2) | 7222 (9.7) | 5959 (9.6) | 3495 (9.1) |
| 3 | 17,888 (10.4) | 11,041 (10.1) | 7253 (9.8) | 5998 (9.7) | 3611 (9.4) |
| 4 | 16,956 (9.8) | 10,546 (9.6) | 7032 (9.5) | 5826 (9.4) | 3377 (8.8) |
| 5 | 18,121 (10.5) | 11,395 (10.4) | 7645 (10.3) | 6347 (10.3) | 3834 (10.0) |
| 6 | 17,887 (10.4) | 11,269 (10.3) | 7535 (10.1) | 6280 (10.1) | 3803 (9.9) |
| 7 | 17,375 (10.1) | 10,999 (10.0) | 7425 (10.0) | 6208 (10.0) | 3900 (10.2) |
| 8 | 17,219 (10.0) | 11,118 (10.1) | 7664 (10.3) | 6382 (10.3) | 4092 (10.7) |
| 9 | 16,763 (9.7) | 11,100 (10.1) | 7936 (10.7) | 6677 (10.8) | 4295 (11.2) |
| 10 (most deprived) | 14,915 (8.6) | 10,544 (9.6) | 7864 (10.6) | 6742 (10.9) | 4685 (12.2) |
| Missing | 206 (0.1) | 136 (0.1) | 89 (0.1) | 69 (0.1) | 59 (0.2) |

MLTC multiple long-term conditions.

[a]Deciles are based on the whole UK Biobank cohort (i.e. also includes people without a continuous primary care record).

## Discussion

Choice of data source has major implications on ascertainment of most conditions, and for many conditions there is poor agreement between data sources. Whilst agreement between data sources is high for many endocrine conditions and neoplasms, it is poor for diseases of the genitourinary system and most mental health conditions. Compared to using all three data sources to identify conditions, primary care records alone identify the highest proportion for 62 of the 80 conditions, compared to 13 conditions for UKB baseline assessment data alone and only five rare conditions for hospital records alone. Primary care records are particularly valuable for ascertaining conditions for which people are not commonly admitted to hospitals or for which the hospital admission period tends to occur earlier in adulthood and thus outside the period of hospital data availability.

Estimates of MLTC prevalence vary greatly depending on the definition of multimorbidity and depending on the data sources used, with estimates based on primary care data alone higher than those based on other data sources alone and with hospital ascertainment of physical–mental multimorbidity particularly deficient. While the characteristics of the population of people identified as having multimorbidity are similar in analyses using primary care records alone and UKB records alone, analyses based on hospital records alone identify a distinct subset of people with multimorbidity. Consistent with that, the patterns for disparities between subgroups are similar in analyses using primary care alone and UKB records alone, whereas there are differences in analyses based on hospital records

alone, notably that men have higher prevalence of multimorbidity than women which is the reverse of all other analyses.

Strengths of the study include systematic identification of 80 LTCs using data from the UKB baseline assessment centre as well as routine primary care and hospital records. We systematically identify and modify existing code-lists for use in UKB, and create new code-lists where required. We transparently report the process for creating new code-lists and make all code-lists and associated materials available for re-use in UKB and other datasets[22,23].

However, the study has a number of limitations. First, prevalence estimates may not be generalisable because UKB's 5.5% response rate means the cohort is somewhat healthier and more affluent than the general population[24]. However, observed patterns of multimorbidity are very likely to be valid[5]. Second, there is no gold-standard diagnosis to evaluate whether people identified as having a condition by any or all data sources actually have that condition[23]. We take steps to reduce both type I (false-positives) and type II (false-negatives) errors, but gold-standard manual review of the full clinical record is not possible. Sharing of code-lists and associated materials is intended to encourage validation of the code-lists by other researchers and topic experts. Third, primary care data are currently only available for a subset of UKB participants. Although almost all UKB participants consent to primary care data linkage, data extraction is not possible from some general practitioner (GP) practice management systems (the most common reason for lack of linked data) or from practices which do not

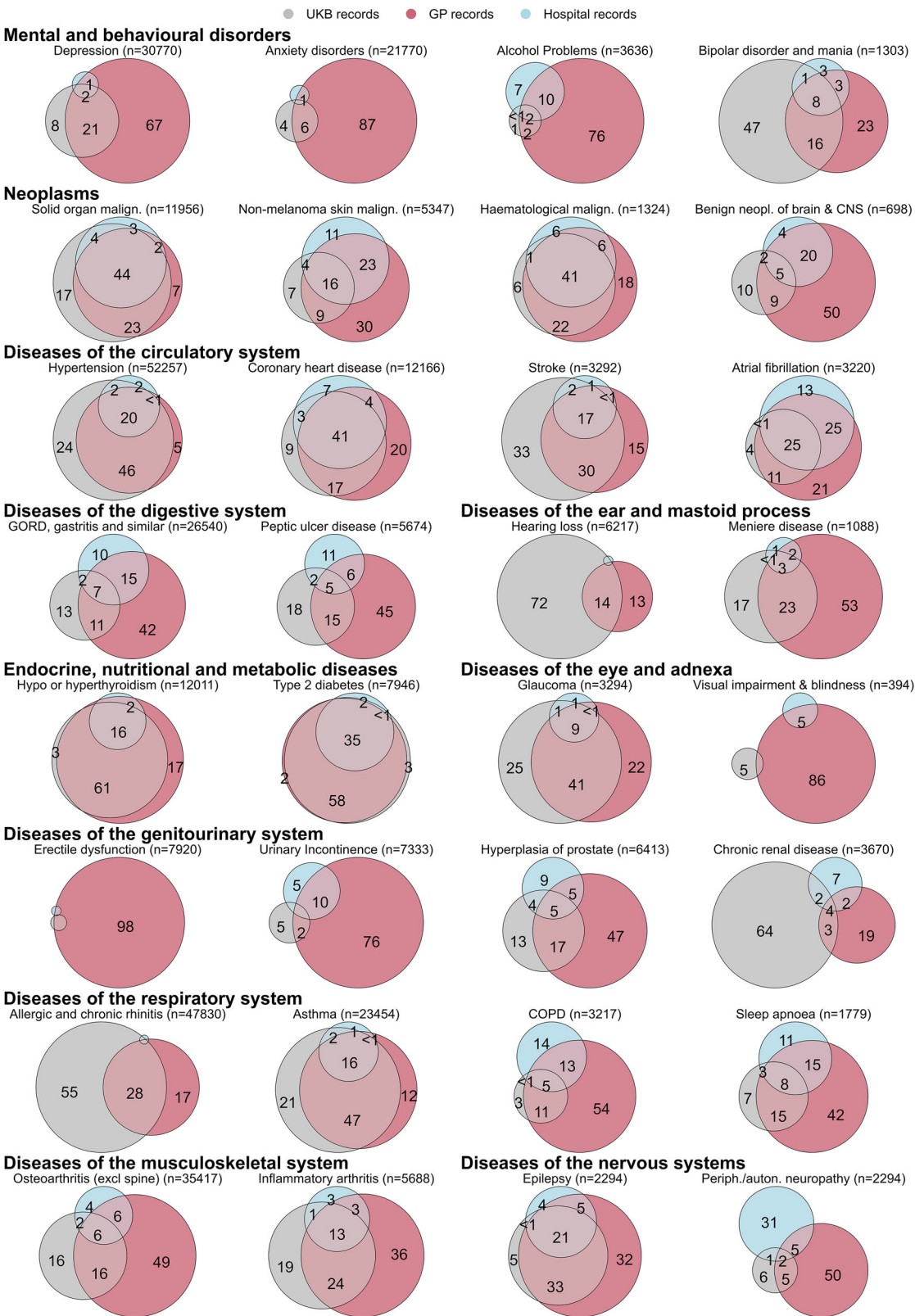

**Fig. 1 | Proportional Venn diagrams of concordance between data sources for selected conditions (using data from all three data sources).** Circle areas and numbers are proportion of records identified. We have suppressed some cells to avoid disclosing small numbers, in keeping with UK Biobank guidelines[48], and to enhance legibility. Proportional Venn diagrams for all conditions by body system are shown in Supplementary Figure 4 (see 'Proportional Venn diagrams'). Auton. autonomic, CNS central nervous system, COPD chronic obstructive pulmonary disease, GORD gastro-oesophageal reflux disorder, neopl neoplasm, periph peripheral.

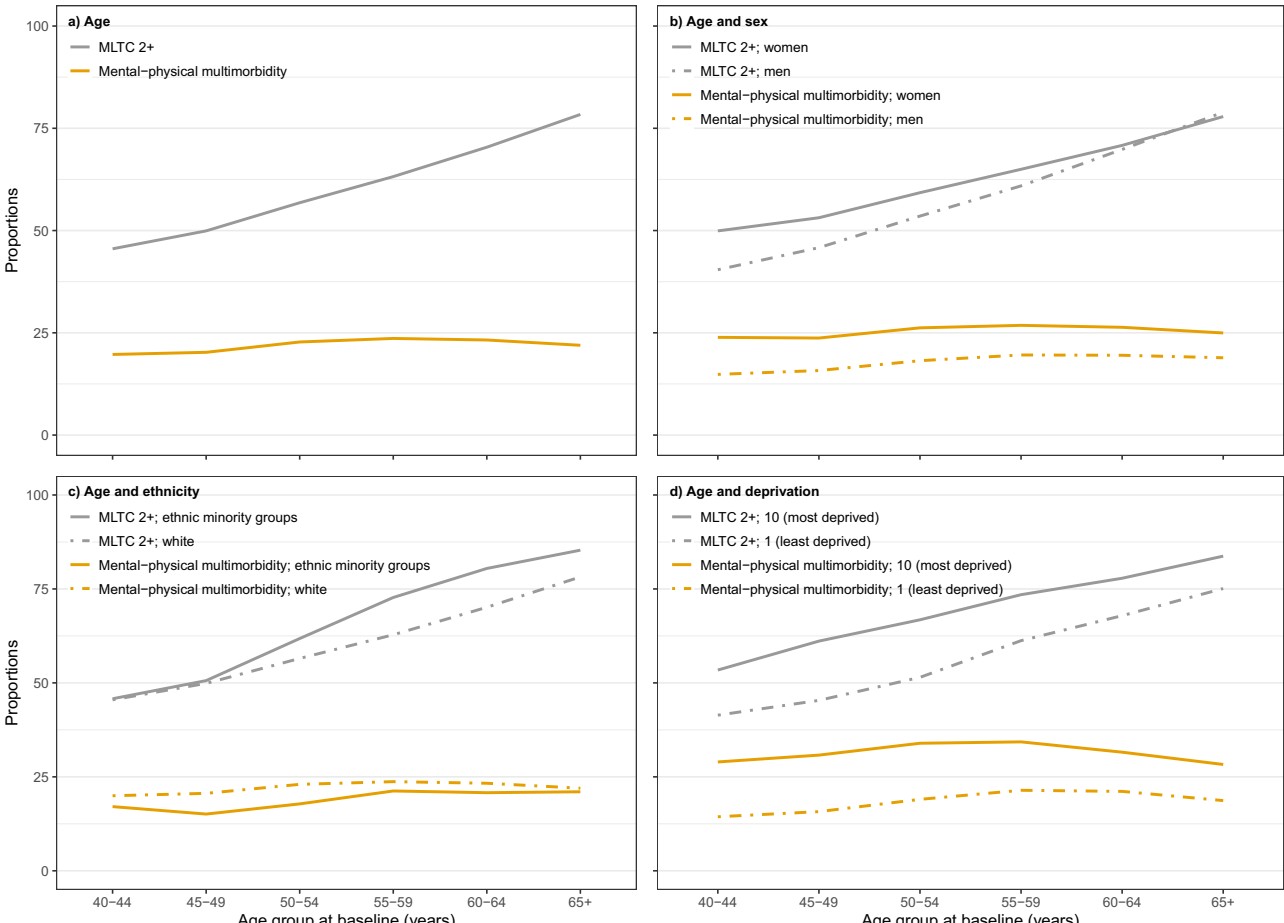

**Fig. 2 | Prevalence of multimorbidity by age, and sex, ethnicity and deprivation for MLTC 2+ and mental–physical multimorbidity[*].** Prevalence of MLTC2+ (grey) and mental–physical multimorbidity (yellow) by age (**a**), sex (**b**), ethnicity (**c**), and deprivation (**d**). *Results for MLTC 3+ and MLTC 3+ from 3+ are shown in Supplementary Fig. 4; MLTC multiple long-term conditions, MLTC 2+ ≥2 long-term conditions (LTCs), MLTC 3+ ≥3 LTCs, MLTC 3+ from 3+ ≥3 LTCs from ≥3 body systems, mental–physical multimorbidity ≥2 LTCs where ≥1 mental health LTC and ≥1 physical health LTC.

consent to extraction[25]. Lack of linkage is therefore driven primarily by choice of GP system rather than individual opt-out. This has implications on the length of follow-up of participants who move to a practice not currently covered in UKB and decreased statistical power. However, we expect the pattern of findings in this study to apply to the whole population. Furthermore, our results will inform decisions about the trade-off between maximising number of participants versus maximising ascertainment of conditions until the expected extension of primary care data linkage to all participants happens. Fourth, hospital and cancer registry data are constrained to an 8-year look-back to ensure consistency across these data sources and countries, but may therefore underestimate hospital recording of conditions which are incident earlier in adulthood (e.g. severe mental health conditions)[26]. Fifth, our findings are specific to the UKB context and the concordance of health conditions among data sources may be different in other countries or among other age groups. However, given that UKB has >30,000 registered research users and thousands of projects led from research organisations around the world, guidance on how to use Read v2 and CTV3 codes and the availability of code-lists is of relevance to a large international community, especially to those who are not used to working with these coding systems. We encourage researchers to apply our systematic methodological approach in other datasets to explore potential differences in findings.

The estimated prevalence of multimorbidity varies greatly across studies due to differences in the definition and measurement of multimorbidity as well as the data sources used[3,8], but our prevalence estimates are similar to the same age groups in a previous study using Clinical Practice Research

Datalink GOLD data and using a similar list of conditions, although physical–mental health multimorbidity prevalence is somewhat higher in that study[20]. Consistent with previous research, our study shows that women and particularly those living in more deprived areas have a higher prevalence of multimorbidity[20,27–29]. However, the size of observed sex and deprivation differences is generally smaller in our study, probably because of selection effects in study recruitment. In five out of seven studies included in a recent systematic review, multimorbidity prevalence is higher among ethnic minorities in the UK than among individuals with a white ethnic background[30]. We find that mental–physical multimorbidity is more common among those with a white ethnic background whereas other multimorbidity definitions are more common among ethnic minority groups. This could be due to differences in health-seeking behaviour (e.g. how 'distress' is conceptualised in different cultures and presented—or not —to healthcare) or diagnostic behaviour (e.g. how clinicians respond to distress and mental health symptoms)[31,32].

Existing UKB publications examining multimorbidity mostly rely on a list of 43 LTCs[29,33], and identify LTCs using data from the UKB baseline assessment, hospital records and death records. Few publications use primary care records to identify LTCs[5,34,35]. Those studies that use primary care records either (1) focus on Read v2 which disregards primary care data that are recorded in CTV3 or (2) use a 'first occurrence' data field created by the UKB team[36], without appreciating that primary care data are currently only available for a subset of the cohort and as such primary care data only contributes to 'first occurrences' for a subset of UKB participants.

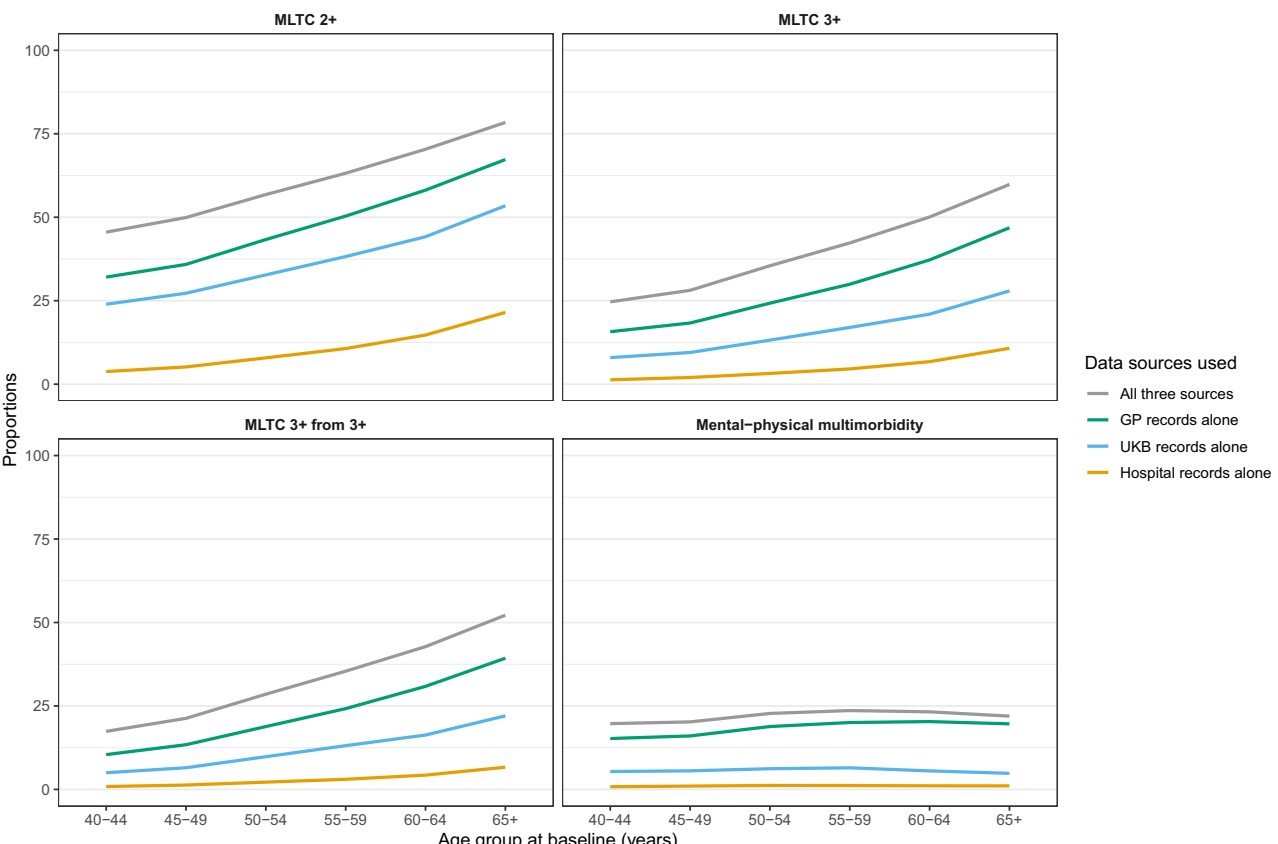

**Fig. 3 | Prevalence of multimorbidity by age using four definitions (MLTC 2+, MLTC 3+, MLTC 3+ from 3+, and mental–physical multimorbidity) and different data sources (GP records, UKB records, and/or hospital records).** Prevalence of MLTC 2+ (top left), MLTC 3+ (top right), MLTC 3+ from 3+ (bottom left), and mental–physical multimorbidity (bottom right), identified using all three data sources (grey), primary care records along (green), UK Biobank records alone (blue), and hospital records alone (yellow); GP general practitioner, MLTC multiple long-term health conditions, MLTC 2+ ≥2 long-term conditions (LTCs), MLTC 3+ ≥3 LTCs; MLTC 3+ from 3+ ≥3 LTCs from ≥3 body systems, mental–physical multimorbidity ≥2 LTCs where ≥1 mental health LTC and ≥1 physical health LTC, UKB UK Biobank.

The poor concordance of both genitourinary and mental health conditions highlights the importance of equal priority being given to mental and physical health care, also known as parity of esteem[37]. There may be several explanations for poor concordance across data sources, one of which is that many conditions with poor concordance, such as mental health conditions, erectile dysfunction and incontinence are stigmatised conditions which have large impacts on quality of life but are often overlooked by physical health services[38,39].

The high prevalence of multimorbidity in middle and early old age highlights the importance of re-orienting models of care towards providing care for people with multimorbidity rather than solely focussing on single diseases, which includes provision of care in multidisciplinary teams and enhanced continuity of care[40]. Our findings highlight that the burden of multimorbidity varies across population subgroups, with higher burden in middle-age in women, more deprived groups and people from ethnic minority groups. This supports repeated calls for a re-distribution of health care provision and funding according to health care needs, in the face of evidence of the persistence of the inverse care law[41,42]. In addition, given the complexity of delivering care to people living with multimorbidity, it is essential to improve recruitment, training and retention of doctors in areas of particularly high health care needs[43]. There are a number of research implications. First, despite code-lists being central to routine data research, the lack of methodological standard for the construction, sharing, revision and reuse of clinical code-lists is recognised as hindering comparability and replication of research[20,21]. The UKB outcome adjudication group developed code-lists for 12 algorithmically-defined health outcomes[19,44,45], and Denaxas and colleagues created phenotyping algorithms for 31

biomarkers[46]. This study extends the list of available CTV3 code-lists to 80 LTCs and provides access to 80 matched Read v2 and ICD-10 code-lists for use in UKB, modified either to be more focused (e.g. more severe hearing impairment) and/or to ensure consistency between Read v2 and CTV3 code-lists.

Second, this study highlights the importance of choice of data source. For example, if the focus of interest are common mental health conditions, then there is a strong argument for using primary care data because hospital data is very partial. This applies more generally to any LTC where there is considerable variation in ascertainment by data source, and we encourage researchers to carefully consider what is right for their particular purpose in their specific context and to be explicit in justifying their choices.

Third, given that the creation of code-lists is very time-consuming, it would also be valuable to explore algorithmic or data-driven approaches to code-list development, especially given the implementation of even larger ontologies, such as SNOMED-CT[47]. However, any new data-driven tools needs to be described in sufficient detail to enable researchers to explore and comment on their functionality in their specific context, and the observed inconsistency in ascertainment highlights the need for more systematic validation of all code-lists[23]. In conclusion, this study extends international consensus recommendations for measuring multimorbidity in research which address condition choice (what to measure)[9], by highlighting that choice of data source has additional major impact on the ascertainment of individual LTCs and of multimorbidity under a range of definitions, with implications for researchers' choices in terms of 'how to measure' multimorbidity. The optimal choice depends on the purpose of a particular

research project, but researchers should explicit state and justify their choices in order to improve reproducibility of research.

## Data availability

All bona fide researchers in academic, commercial, and charitable settings can apply to use the UK Biobank resource for health-related research in the public interest (www.ukbiobank.ac.uk/register-apply/). As part of the application for UK Biobank data, researchers can ask for information stored in participants' electronic health records, which is then provided to researchers by UK Biobank. The source data for Fig. 1 and Supplementary Fig. 4 is in Supplementary Data 5. The source data for Fig. 2 and Supplementary Fig. 8 is in Supplementary Data 6. The source data for Fig. 3 is in Supplementary Data 7. Any additional requests for information can be directed to, and will be fulfilled by, the corresponding authors.

## Code availability

Code-lists for all long-term conditions included in this paper have been deposited in the GitHub repository 'mltc-codelists' (version 1.0.0.) and are available for other researchers on https://github.com/rprigge-uoe/mltc-codelists.git[14].

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

## Acknowledgements

This work was co-funded by the Medical Research Council and the National Institute for Health Research through grant number MR/S028013/1. The study was conducted using the UK Biobank Resource under application number 57213. The funders of the study had no role in study design, data collection, analysis or interpretation, or writing of the report. The authors would like to thank the UK Biobank participants and the UK Biobank staff for their contributions to this study. The authors would also like to thank: Pat Watson, a public member of our advisory board, for providing thoughtful feedback throughout our project; Dr Emma Davidson for providing us with unpublished depression code-lists; Dr Kristiina Rannikmäe for providing us with stroke, B12 deficiency anaemia, folate deficiency anaemia and iron deficiency anaemia that were unpublished at the time; Dr Clare MacRae and Dr Eleojo Abubakar for their guidance on creating proportional Venn diagrams in R.

## Author contributions

B.G., C.A.J., S.W.M., D.R.M., J.D.N., D.J.S. and C.S. conceived the study and acquired funding for it. R.P. and K.J.F. accessed and verified the underlying data reported in the manuscript and analysed the data. R.P. conducted the literature search and wrote the original draft of the manuscript. B.G., C.A.J., S.W.M., D.R.M., J.D.N., D.J.S., C.S., R.P., K.J.F. and P.A.T.K. reviewed and edited the manuscript and approved the final version.

## Competing interests

The authors declare no competing interests.
