## [Peer Review file · Communications Medicine]

Robustly measuring multimorbidity using disparate linked datasets

Corresponding Author: Dr Regina Prigge

Version 0:

Reviewer comments:

Reviewer #3

(Remarks to the Author)

This paper by Prigge et al seeks to compare how well three different data sources capture long-term conditions and in turn impact the reported prevalence of multimorbidity. Overall, I think this is a well conducted study, and uses rigorous methods. The UK biobank, which is linked to the hospital, cancer, and primary care records provides an ideal dataset to study this, as comparisons can be made within the same person. The tables and figures provided a concise presentation of results, and the supplementary material provided needed supporting information. The analytic methods are simple, but appropriate for the aim of the study. This is an important, but I think under-appreciated topic - how much the choice of data source can impact the amount of multimorbidity detected in a database. I think the paper can have a substantial impact on the field.

I only have a few additional comments:

1) I am confused why creatine measures were used to define CKD in UK biobank, while all other UK biobank measures are self-report through touchscreen survey or nurse interview. I think it would be better to be consistent and just use the nurse interview response. The paper is framed as a comparison of three data source types: hospital/cancer records, primary care records, and survey self-report. Using lab values for this one LTC deviates from this.

2) Related, I am not sure we can say <60 eGFR is a diagnosis of CKD. I am not a clinician, but I thought a reading this low was just suggestive of possible CKD. CKD rates were much higher in UK biobank than other sources, which was surprising for such a serious condition. This makes me think many of these are not true cases.

3) Methods: it should be stated in the main body of the manuscript how LTCs are assessed in the UK biobank baseline assessments. The current write up focuses only on the administrative data codes.

4) Some discussion about how these results may or may not apply in other data sources outside the UKB is warranted. For example, I don't think US primary care clinics use Read v2 or CTV3 codes, but rather ICD-10 codes. Further many primary care practices are affiliated with health systems and use their EHR, so the differences between primary care and hospital records may not be that different.

5) The authors should be commended for providing the codes needed to reproduce this work on github, which will be a resource to other researchers that wish to use this code in their research.

Reviewer #4

(Remarks to the Author)

This is an excellent paper in an area that is ripe for publication: the effort toward greater standardization and consensus around multimorbidity/MCC/MLTC. I commend the author team for their comprehensive assessment of multiple data sources, the investigation of their robustness in identifying specific long term chronic conditions, and their presentation of data visualizations of their findings. That said, there are a few clarifications and refinements to the manuscript that I would urge the authors to consider in earnest to strengthen their work:

1. In the Introduction, the authors make a compelling case regarding (line 61) "which LTCs to measure." I think this might be a misnomer. Their contribution will be on which LTCs to include in MM/MLTCs measures and data source considerations to those inclusion decisions. It is imperative to be clear about what the current issues and concerns are in this field and the insight this particular paper will provide with the many gaps in this field. Citation of the following papers would contribute to and supplement these arguments:

- Suls J, Bayliss EA, Berry J, et al. Measuring Multimorbidity: Selecting the Right Instrument for the Purpose and the Data Source. *Med Care*. 2021;59(8):743-756. doi:10.1097/MLR.0000000000001566

2. Line 91: the authors do not provide sufficient rationale for using 8 years of lookback in registry and hospital records. Brief rationale to orient the reader on why 8 years was selected (e.g., is this what your team was limited to--max lookback horizon?), or were considerations of aligning time periods with other data sources (ie, the GP and UKB data time frames?). It is unclear in the manuscript text.

3. Lines 95-100: It appears the authors whittled down the 308 conditions to 80 with guidance from the Ho et. al. Delphi consensus study, but the reader is left wondering what criteria were applied to arrive at the 80 conditions. A brief description here in the manuscript, and/or more lengthy details provided in the appendix regarding the decisionmaking processes and criteria applied is warranted particularly given the need to provide clear guidance to the field on how they arrived at this list of 80.

4. The authors make a good case as to why patients may and may not be captured in the various data sources included in this study. Hospital records are generated, by definition, for patients with uncontrolled, exacerbated, and severe LTC, or for patients who may have LTC that are under control but are seeking urgent and inpatient care for acute trauma. This means these patients are likely a subset of patients seen in general practice (the GC data set). This exposition of the varying, and likely "nested" denominators, is not very well discussed or laid out in the paper. It could be more clearly presented in the methods and elaborated upon in the discussion.

5. Related to point 4, given these different "subsets" of patients and individual capture in the three data sources examined, it would be good to present the results with these constraints in mind (lines 178-192). Specifically, I wouldn't make much hay that there is such little overlap between the 3 data sources (authors report a median of 4.7% for conditions reported in all 3). I would find it interesting to report and discuss overlap between the GP and UKB data sources and focus on identification of specific conditions in those two data sources in particular.

6. As a continuation to point 5, I strongly suggest the authors consider a data visualization to display the concordance between the GP and UKB data for the LTC under consideration. I would be very interested in seeing something like a tornado plot (similar to what is presented in this paper: Voss RW, Schmidt TD, Weiskopf N, et al. Comparing ascertainment of chronic condition status with problem lists versus encounter diagnoses from electronic health records. *J Am Med Inform Assoc*. 2022;29(5):770-778. doi:10.1093/jamia/ocac016) that details ascertainment between GP and UKB.

Reviewer #5

(Remarks to the Author)

Thank you for the opportunity to review this interesting manuscript. It is a highly topical and timely article that examines how the choice of data source impacts estimated prevalence of 80 individual long-term conditions (LTCs) and multimorbidity. It is a much-needed paper that has the potential to transform how multimorbidity research is conducted. I note that the study has been undertaken by senior international experts in multimorbidity. They show that the choice of data source can result in substantial variation in condition inclusion and thus a priori rationale for MLTC condition lists must be clearly justified.

The paper was well-written and easy to follow. The key message is clear and the results add substantially to the existing literature. Some minor points below for the authors to consider that may strengthen the paper:

1. Could they consider adding further justification for including these four specific data sources: (1) UKB (2) Hospital admission records coded with ICD-10 for diagnoses and OPCS-4 for procedures (3) Cancer registry; and (4) Primary care records. Were these databases available to the authors ie- convenience sampling or was there a methodological justification for using these specific databases? In particular why UKB? Might different data sources have produced different findings and could the implication of this be expanded
2. Further consideration is needed on the implication of having only half of the participants with linkage to primary care records
3. Additional text or supplementary material is needed on the process used by the three clinicians (BG, SM, DS) who selected LTCs to include from a previously published set of 308 acute and chronic conditions. The designation and background of these clinicians (primary vs secondary vs interest in particular conditions) may well have impacted choice of condition inclusion, and thus findings.
4. I note the sample restriction of middle-aged adults in UKB. Multimorbidity more frequently impacts older adults with increasing morbidity in older age. Is this therefore a substantial limitation of the dataset used? Could the authors expand on implications if the sample was more diverse in age and reflective of the real world?
5. Could the authors consider potential explanations for the poor concordance in both genitourinary and mental health conditions; and then highlight more specific actions to deal with this in future work for researchers/clinicians?

If the above points are revised, this will be a robust manuscript. I really enjoyed reading this and hope to see it in press soon.

Version 1:

Reviewer comments:

Reviewer #1

(Remarks to the Author)

The author's revisions have significantly improved the manuscript, and they were responsive to my comments from the past review. Therefore, I have no further comments.

I read through each of the Reviewer #3 comments and responses. In my opinion the author's did a good job of responding to the reviewer comments, and adding appropriate discussion points to the manuscript.

Reviewer #2

(Remarks to the Author)

The authors have been very responsive to comments and critiques. I particularly appreciated the addition of tornado plots to the appendices. They provide great additional information. I did not intend for these to replace or supplant the Venns, on this the authors and I agree with placing them among the supplemental information for readers to refer to. Thank you, and I have no further comments to add.

Reviewer #1:

This paper by Prigge et al seeks to compare how well three different data sources capture long-term conditions and in turn impact the reported prevalence of multimorbidity. Overall, I think this is a well conducted study, and uses rigorous methods. The UK biobank, which is linked to the hospital, cancer, and primary care records provides an ideal dataset to study this, as comparisons can be made within the same person. The tables and figures provided a concise presentation of results, and the supplementary material provided needed supporting information. The analytic methods are simple, but appropriate for the aim of the study. This is an important, but I think under-appreciated topic - how much the choice of data source can impact the amount of multimorbidity detected in a database. I think the paper can have a substantial impact on the field.

I only have a few additional comments:

1) I am confused why creatine measures were used to define CKD in UK biobank, while all other UK biobank measures are self-report through touchscreen survey or nurse interview. I think it would be better to be consistent and just use the nurse interview response. The paper is framed as a comparison of three data source types: hospital/cancer records, primary care records, and survey self-report. Using lab values for this one LTC deviates from this.

We appreciate that the ascertainment of CKD in UK Biobank baseline assessment data stands out from the assessment of other conditions in this data source. Your comment highlights that there is often potential to define diseases in various different ways, which matters more for some conditions than others. We used creatinine lab values because CKD whose presence is only measured by reduced eGFR (with no specific renal diagnosis and no proteinuria) is often not communicated to the patient (Stewart et al, 2024), reflecting debate among clinicians whether CKD in this sense is a disease or an attribute of ageing (Guppy et al, 2022). We therefore judged that ascertainment of chronic kidney disease via self-report alone would not be accurate. Unfortunately, we no longer have access to the data so cannot run a sensitivity analysis to assess the effect of our decision on the result of our analysis. However, researchers using UK Biobank can of course run sensitivity analyses to assess the effects of different decisions on their analyses. To be more explicit about the type of information used from the UK Biobank baseline assessment centre, we have added the following sentences to the manuscript (p4):

“UKB baseline assessment data included information from a touchscreen questionnaire, nurse interview, physical and functional measures, and collection of blood, urine, and saliva⁴. We identified relevant information on the presence of 72 conditions, which was largely based on self-report in the touchscreen questionnaire and nurse interview but also included laboratory values of creatinine to ascertain chronic kidney disease (Supplementary Appendix 1).”

In addition, we checked that we refer to this data source as “UKB records” throughout our manuscript (and thus did not mis-label it as “self-report”).

Guppy, M., Glasziou, P., Beller, E., Flavel, R., Shaw, J. E., Barr, E., & Doust, J. (2022). Kidney trajectory charts to assist general practitioners in the assessment of patients with reduced kidney function: a randomised vignette study. *BMJ evidence-based medicine*, 27(5), 288–295. <https://doi.org/10.1136/bmjebm-2021-111767>

Stewart, S., Kalra, P. A., Blakeman, T., Kontopantelis, E., Cranmer-Gordon, H., & Sinha, S. (2024). Chronic kidney disease: detect, diagnose, disclose-a UK primary care perspective of barriers and

enablers to effective kidney care. *BMC medicine*, 22(1), 331. <https://doi.org/10.1186/s12916-024-03555-0>

2) Related, I am not sure we can say <60 eGFR is a diagnosis of CKD. I am not a clinician, but I thought a reading this low was just suggestive of possible CKD. CKD rates were much higher in UK biobank than other sources, which was surprising for such a serious condition. This makes me think many of these are not true cases.

The chosen cut-off of <60 eGFR is based on the Kidney Disease: Improving Global Outcomes (KDIGO) international guidelines (see: <https://kdigo.org/guidelines/ckd-evaluation-and-management/>), and is very commonly used to define CKD in routine data. To make our rationale for choosing this cut-off more explicit, we have added a citation of the guideline to the section in the supplement.

3) Methods: it should be stated in the main body of the manuscript how LTCs are assessed in the UK biobank baseline assessments. The current write up focuses only on the administrative data codes.

As described in our response to your first comment, we have added additional information about this to the manuscript (p4) and have cited an earlier descriptive paper providing additional information (Sudlow et al, 2015).

4) Some discussion about how these results may or may not apply in other data sources outside the UKB is warranted. For example, I don't think US primary care clinics use Read v2 or CTV3 codes, but rather ICD-10 codes. Further many primary care practices are affiliated with health systems and use their EHR, so the differences between primary care and hospital records may not be that different.

We agree that the concordance between data sources may be different in other contexts which re-emphasizes our argument that researchers should carefully consider what is right for their particular purpose and to explicitly justify their choices. We have re-worded the sentence as follows to include a reference to the context the researchers are working in:

"This applies more generally to any LTC where there is considerable variation in ascertainment by data source, and we encourage researchers to carefully consider what is right for their particular purposes in their specific context and to be explicit in justifying their choices." (p13)

In addition, we have added the following statement to the discussion section (p11):

"Fifth, our findings are specific to the UKB context and the concordance of health conditions among data sources may be different in other countries or among other age groups. However, given that UK Biobank has >30,000 registered research users and thousands of projects led from research organizations around the world, guidance on how to use Read v2 and CTV3 codes and the availability of code-lists is of relevance to a large international community, especially to those who are not used to working with these coding systems. We encourage researchers to apply our systematic methodological approach in other datasets to explore potential differences in findings."

5) The authors should be commended for providing the codes needed to reproduce this work on github, which will be a resource to other researchers that wish to use this code in their research.

Many thanks for the positive feedback.

Reviewer #2:

This is an excellent paper in an area that is ripe for publication: the effort toward greater standardization and consensus around multimorbidity/MCC/MLTC. I commend the author team for their comprehensive assessment of multiple data sources, the investigation of their robustness in identifying specific long term chronic conditions, and their presentation of data visualizations of their findings. That said, there are a few clarifications and refinements to the manuscript that I would urge the authors to consider in earnest to strengthen their work:

1. In the Introduction, the authors make a compelling case regarding (line 61) "which LTCs to measure." I think this might be a misnomer. Their contribution will be on which LTCs to include in MM/MLTCs measures and data source considerations to those inclusion decisions. It is imperative to be clear about what the current issues and concerns are in this field and the insight this particular paper will provide with the many gaps in this field. Citation of the following papers would contribute to and supplement these arguments:

**- Suls J, Bayliss EA, Berry J, et al. Measuring Multimorbidity: Selecting the Right Instrument for the Purpose and the Data Source. Med Care. 2021;59(8):743-756.
doi:10.1097/MLR.0000000000001566**

We agree with the reviewer and have rephrased the question in the introduction to: "*Which long-term conditions (LTCs) to include in multimorbidity measures?*" (p2). In addition, we appreciate the suggestion to include the paper by Suls et al (2021) which we have added to our paper as reference 10 (p2). The paper aligns with our manuscript in that it highlights advantages and disadvantages of different data sources for measuring conditions and its conclusion that the choice of the approach to measure multimorbidity should match its purpose.

2. Line 91: the authors do not provide sufficient rationale for using 8 years of lookback in registry and hospital records. Brief rationale to orient the reader on why 8 years was selected (e.g., is this what your team was limited to--max lookback horizon?), or were considerations of aligning time periods with other data sources (ie, the GP and UKB data time frames?). It is unclear in the manuscript text.

We have added a sentence to the methods section explaining that the eight-year lookback period was driven by the minimum length of records available across all secondary data sources and countries (England, Scotland and Wales):

"Cancer registry and hospital records were available from different dates for England, Wales and Scotland, with a minimum of eight years of records prior to the baseline assessments (Supplementary Table 1). To maximise the use of available data whilst ensuring consistency across the secondary care data sources and countries, we therefore defined conditions at baseline using cancer registry and hospital records from the eight years up to and including their baseline assessment date." (p3)

In addition, we have moved our cross-reference to Supplementary Table 1 which outlines the data availability for all data sources and countries (p3).

3. Lines 95-100: It appears the authors whittled down the 308 conditions to 80 with guidance

from the Ho et. al. Delphi consensus study, but the reader is left wondering what criteria were applied to arrive at the 80 conditions. A brief description here in the manuscript, and/or more lengthy details provided in the appendix regarding the decision-making processes and criteria applied is warranted particularly given the need to provide clear guidance to the field on how they arrived at this list of 80.

We have re-worded the paragraph in the main manuscript to enhance clarity regarding the instructions given to the three clinicians who selected the conditions to be used in our project (p4). In addition, we have added a more detailed description of the process to the supplement (see Supplementary Appendix 2) which includes information on the background of the clinicians and their research interest as well as a more detailed description of steps involved in reaching the clinical consensus of including the 78 conditions (plus the two additional conditions based on the Delphi study).

4. The authors make a good case as to why patients may and may not be captured in the various data sources included in this study. Hospital records are generated, by definition, for patients with uncontrolled, exacerbated, and severe LTC, or for patients who may have LTC that are under control but are seeking urgent and inpatient care for acute trauma. This means these patients are likely a subset of patients seen in general practice (the GC data set). This exposition of the varying, and likely "nested" denominators, is not very well discussed or laid out in the paper. It could be more clearly presented in the methods and elaborated upon in the discussion.

We agree that in a perfect world one would expect that patients treated in hospitals are a subset of patients seen in general practice. Whilst this was true for some conditions in our analysis, our Venn diagrams show that some patients are diagnosed with conditions in hospital records but not in GP records. For example, patients with hospital records of conditions of the endocrine, nutritional and metabolic system and neoplasms were largely a subset of patients seen in general practice, whereas this pattern was less evident for diseases of the circulatory system. In keeping with your recommendation below, we have created tornado plots to show the concordance between two data sources at a time, which further illustrates this point visually in the supplement of the paper (Supplementary Figures 2 to 4).

5. Related to point 4, given these different "subsets" of patients and individual capture in the three data sources examined, it would be good to present the results with these constraints in mind (lines 178-192). Specifically, I wouldn't make much hay that there is such little overlap between the 3 data sources (authors report a median of 4.7% for conditions reported in all 3). I would find it interesting to report and discuss overlap between the GP and UKB data sources and focus on identification of specific conditions in those two data sources in particular.

We believe that our Venn diagrams and the concordance between the three data sources is of importance given how extremely widely HES data have been used in combination with UK Biobank baseline assessment centre data by UKB researchers. The Venn diagrams put an emphasis on the proportion of records that is missed in many existing studies focusing solely on UKB and hospital records. However, we understand that the concordance between two data sources can be of interest so have added the following sentence to our results section (p8):

"Focusing on only two data sources at a time, a median of 22.9% (IQR 8.3 – 41.7) of individuals with a condition were identified by primary care and UKB records, 17.3% (IQR 6.5 – 26.7) by primary care

and hospital records and 10.8% (IQR 5.2 – 23.7) by UKB and hospital records (Supplementary Figures 2 to 4).”

6. As a continuation to point 5, I strongly suggest the authors consider a data visualization to display the concordance between the GP and UKB data for the LTC under consideration. I would be very interested in seeing something like a tornado plot (similar to what is presented in this paper: Voss RW, Schmidt TD, Weiskopf N, et al. Comparing ascertainment of chronic condition status with problem lists versus encounter diagnoses from electronic health records. *J Am Med Inform Assoc.* 2022;29(5):770-778. doi:10.1093/jamia/ocac016) that details ascertainment between GP and UKB.

As described above, we have added tornado plots showing the overlap between two data sources at a time to the supplement of the paper (see Supplementary Figures 2 to 4). In our opinion, there is great value in showing Venn diagrams of the concordance between the three data sources in the main manuscript 1) given the extensive use of UKB and hospital records in existing research using UK Biobank data and 2) to demonstrate the value of using primary care records and 3) to highlight the complexities of the overlap between the data sources and the careful consideration needed in deciding which data source to use for projects with an interest in individual long-term conditions or multimorbidity.

Reviewer #3:

Thank you for the opportunity to review this interesting manuscript. It is a highly topical and timely article that examines how the choice of data source impacts estimated prevalence of 80 individual long-term conditions (LTCs) and multimorbidity. It is a much-needed paper that has the potential to transform how multimorbidity research is conducted. I note that the study has been undertaken by senior international experts in multimorbidity. They show that the choice of data source can result in substantial variation in condition inclusion and thus a priori rationale for MLTC condition lists must be clearly justified.

The paper was well-written and easy to follow. The key message is clear and the results add substantially to the existing literature. Some minor points below for the authors to consider that may strengthen the paper:

1. Could they consider adding further justification for including these four specific data sources: (1) UKB (2) Hospital admission records coded with ICD-10 for diagnoses and OPCS-4 for procedures (3) Cancer registry; and (4) Primary care records. Were these databases available to the authors ie- convenience sampling or was there a methodological justification for using these specific databases? In particular why UKB? Might different data sources have produced different findings and could the implication of this be expanded

This work was part of a larger project co-funded by the Medical Research Council and the National Institute for Health Research focused on furthering our understanding of the relationship between depression and trajectories of physical multimorbidity accrual in UK Biobank. Thus, to some extent this was a convenience sample of data available to us as researchers. However, it was also restricted by design to the data that UK Biobank had access with the aim of linking to routinely collected national health datasets that enable as comprehensive as possible coverage of health

conditions. We have added the following statement to the discussion section to comment on the generalizability of findings in other contexts (p11):

“Fifth, our findings are specific to the UKB context and the concordance of health conditions among data sources may be different in other countries or among other age groups. However, given that UK Biobank has >30,000 registered research users and thousands of projects led from research organizations around the world, guidance on how to use Read v2 and CTV3 codes and the availability of code-lists is of relevance to a large international community, especially to those who are not used to working with these coding systems. We encourage researchers to apply our systematic methodological approach in other datasets to explore potential differences in findings.”

2. Further consideration is needed on the implication of having only half of the participants with linkage to primary care records

We have now clarified in our Methods section that *“Linked primary care data was available from Scotland, Wales and practices in England that used either the TPP or Vision practice management systems. We excluded primary care records from the UKB extract of the Vision practice management system in England because this linked dataset did not include participants who died before data extraction.^{11”} (p3).*

Furthermore, we have expanded on this point in our discussion section, as follows (p10):

“Although almost all UK Biobank participants consent to primary care data linkage, data extraction is not possible from some GP practice management systems (the most common reason for lack of linked data) or from practices which do not consent to extraction.²⁴ Lack of linkage is therefore driven primarily by choice of GP system rather than individual opt-out. This has implications on the length of follow-up of participants who move to a practice not currently covered in UK Biobank and decreased statistical power. However, we would expect the pattern of findings in this study to apply to the whole population.”

3. Additional text or supplementary material is needed on the process used by the three clinicians (BG, SM, DS) who selected LTCs to include from a previously published set of 308 acute and chronic conditions. The designation and background of these clinicians (primary vs secondary vs interest in particular conditions) may well have impacted choice of condition inclusion, and thus findings.

We have re-worded the paragraph in the main manuscript to enhance clarity regarding the instructions given to the three clinicians who selected the conditions to be used in our project (p4). In addition, we have added a more detailed description of the process to the supplement (see Supplementary Appendix 2) which includes information on the background of the clinicians and their research interest as well as a more detailed description of steps involved in reaching the clinical consensus of including the 78 conditions (plus the two additional conditions based on the Delphi study).

4. I note the sample restriction of middle-aged adults in UKB. Multimorbidity more frequently impacts older adults with increasing morbidity in older age. Is this therefore a substantial limitation of the dataset used? Could the authors expand on implications if the sample was more diverse in age and reflective of the real world?

As highlighted in our response to your first comments, we added a paragraph to the discussion section to comment on the generalizability of findings in other samples/ other contexts (p11).

5. Could the authors consider potential explanations for the poor concordance in both genitourinary and mental health conditions; and then highlight more specific actions to deal with this in future work for researchers/clinicians?

There are several explanations of the poor concordance in both genitourinary and mental health conditions. For mental health conditions this is likely a mix of (1) Most admissions are for physical problems so mental health conditions aren't commonly the primary reason for admission; (2) Since younger people are rarely admitted to hospital, but commonly have mental health problems, that will also contribute; (3) Mental health recording will often be mentioned as an aside, but mental health is stigmatized and often overlooked by physical disease services, so is probably less likely to be recorded during a physical condition admission than other physical conditions. For genitourinary conditions it is important to look at what is in the group. The least recorded in the hospital are erectile dysfunction and incontinence, both stigmatized conditions which are often overlooked by services (not asked about, not taken that seriously).

We have added text about an important implication of these potential explanations of our findings in our manuscript (p12):

“The poor concordance of both genitourinary and mental health conditions highlights the importance of equal priority being given to mental and physical health care, also known as “parity of esteem”³⁶. There may be several explanations for poor concordance across data sources, one of which is that many conditions with poor concordance, such as mental health conditions, erectile dysfunction and incontinence are stigmatised conditions which have large impacts on quality of life but are often overlooked by physical health services^{37,38}.”